# Quality Evaluation of Dietary Supplements for Weight Loss Based on *Garcinia cambogia*

**DOI:** 10.3390/nu14153077

**Published:** 2022-07-27

**Authors:** Adal Mena-García, Angie Julieth Bellaizac-Riascos, Maite Rada-Mendoza, Diana María Chito-Trujillo, Ana Isabel Ruiz-Matute, María Luz Sanz

**Affiliations:** 1Instituto de Química Orgánica General (CSIC), Juan de la Cierva 3, 28006 Madrid, Spain; a.mena@iqog.csic.es (A.M.-G.); ana.ruiz@csic.es (A.I.R.-M.); 2Grupo de Investigación Biotecnología, Calidad Medioambiental y Seguridad Agroalimentaria (BICAMSA), Universidad del Cauca, Popayán 190003, Colombia; angieb@unicauca.edu.co (A.J.B.-R.); mrada@unicauca.edu.co (M.R.-M.); dchito@unicauca.edu.co (D.M.C.-T.)

**Keywords:** *Garcinia gummi-gutta*, *Garcinia cambogia*, gas chromatography–mass spectrometry, liquid chromatography, (-)-hydroxycitric acid, food supplements

## Abstract

Food supplements of plant origin for weight control are increasingly being demanded by consumers as a way to promote good health. Among them, those based on *Garcinia cambogia* (GCFS) are widely commercialized considering their bioactive properties, mainly due to (-)-hydroxycitric acid ((-)-HCA). However, recently, controversy has arisen over their safety; thus, further research and continuous monitoring of their composition is required. Hence, in this work, a multi-analytical approach was followed to determine not only (-)-HCA but also other constituents of 18 GCFS, which could be used as quality markers to detect fraudulent practices in these samples. Discrepancies between the declared (-)-HCA content and that experimentally determined were detected by LC–UV in 33% of the samples. Moreover, GC–MS analyses of GCFS allowed the detection of different compounds not present in *G. cambogia* fruits and not declared on supplement labels, probably related to heat exposure or to the addition of excipients or other extracts. This multi-analytical methodology is shown to be advantageous to address different fraudulent practices affecting the quality of these supplements.

## 1. Introduction

Obesity is among the diseases with the highest incidence in current times, especially in developed countries. In 2016, more than 39% of the world’s adult population was overweight, and 13% were obese [1]. An adequate and balanced diet provides all the necessary nutrients for the development and maintenance of a healthy body. However, in developed countries there is a growing trend towards the consumption of food supplements to increase the intake of some nutrients or bioactive compounds and, thus, to achieve an added physiological effect. Among them, food supplements for overweight control (FSOC) of plant origin are in high demand, being perceived by consumers as a non-harmful or natural alternative to prevent or treat diseases related to obesity, as well as to promote, in general, a good state of health [2]. In addition, it is described that the use of some of these products provides advantages over compounds of synthetic origin, due to the synergistic effect of their components or the fewer side effects they produce [3,4].

Currently, the commercialization of FSOC in pharmacies, herbalists, specialized stores, or supermarkets is being complemented by their growing acquisition via the Internet [2]. The advantages for the consumer in terms of speed, easy accessibility, and wide availability of products have led to a large increase in the volume of business associated with their online trade [5]. However, the quality of these commercial products is not always well established, and they are often subject to potential labeling fraud. In addition, FSOC have been the subject of food alerts, which have led to reformulations, partial withdrawals from the market, or restrictions on their distribution [2,5].

*Garcinia gummi-gutta* is a plant of Asian origin, widely known by its old scientific name, *Garcinia cambogia* (this term is used in this manuscript due to its greater acceptance). *G. cambogia* fruit rind, previously dried and smoked (named as kudam puli), is traditionally used in cooking, and its extracts are commercialized as FSOC considering the bioactive properties of some of their constituents, mainly of (-)-hydroxycitric acid [(-)-HCA] [6]. Commercial food supplements based on *G. cambogia* (GCFS) contain between 50 and 60% (-)-HCA and are one of the supplements most widely marketed worldwide [7,8]. (-)-HCA is a metabolic regulator of obesity and lipid abnormalities in mammalian systems. It inhibits the enzyme citric acid lyase, which is required in the synthesis of fatty acids, and reduces the body’s ability to form adipose tissue [9,10,11]. Along with its weight control properties, various other benefits have also been attributed to this bioactive present in *G. cambogia*, such as its anti-inflammatory, neuroprotective, antidiabetic, antioxidant, and antimicrobial activities [12,13,14]. However, certain clinical and in vivo studies have indicated that the activity of (-)-HCA as an inducer of weight loss is limited or even null and its consumption could cause nausea and headache, among other side effects [5,10,15]. Moreover, controversy has arisen in recent years over the safety of *G. cambogia* supplements [10,15]. Acute liver failure could potentially be associated with the intake of *G. cambogia* extracts for weight loss [10,16,17], raising an important concern regarding its safety. Nevertheless, the mechanism of (-)-HCA toxicity in the liver is not clearly defined, and the potential toxic amount of (-)-HCA has not been determined. In this context, continuous and strict monitoring of GCFS should be promoted to evaluate the potential health risks that may be useful for regulatory authorities, and analytical tools to estimate their whole composition are needed.

Although most studies point at (-)-HCA as the main component responsible for the potential toxic effects of GCFS, it should be noted that the *G. cambogia* fruit (which contains high concentrations of this acid) has been commonly consumed in Asia for centuries, being considered as safe [8]. It is worth noting that *G. cambogia* supplements are also constituted by many other components that have been scarcely studied, so the toxicity cannot be exclusively and reliably attributed to the natural source [15].

Chromatographic techniques such as liquid chromatography (LC) and gas chromatography (GC) are powerful techniques for the qualitative and quantitative analysis of complex matrices such as vegetal extracts and food supplements [18,19]. Some works have focused on the determination of (-)-HCA in *G. cambogia* fruits by LC with ultraviolet (UV) detection [20,21,22,23] or GC coupled to mass spectrometry (MS) [24]. However, studies regarding the characterization of GCFS aimed to evaluate their quality are very limited and only focused on the determination of the (-)-HCA content [21,25]. The characterization of other constituents of GCFS, which could be used as quality markers, and the detection of additional fraudulent practices have not been previously addressed in these samples.

Therefore, in this manuscript, a multi-analytical approach was followed to determine the quality of *G. cambogia* food supplements. LC–UV and GC–MS were used for the qualitative and quantitative analysis of (-)-HCA, as well as other components of these supplements that could affect their quality.

## 2. Materials and Methods

### 2.1. Reagents and Standards

Analytical standards of arabinose, arabitol, L-ascorbic acid, citric acid, fructose, glucose, galactose, (-)-calcium hydroxycitrate tribasic [(-)-HCA], (-)-hydroxycitric acid lactone [(-)-HCAL], *muco*-inositol, *myo*-inositol, maltose, maltotriose, maltotetraose, mannitol, pinitol, phenyl-*β*-glucoside (internal standard), raffinose, sucrose, and verbascose were obtained from Sigma Aldrich (St. Louis, MO, USA). Derivatization reagents including hydroxylamine chloride, anhydrous pyridine, hexamethyldisilazane (HMDS), and trifluoroacetic acid (TFA) were also acquired from Sigma Aldrich (St. Louis, MO, USA).

### 2.2. Samples

Dried and smoked fruit rinds from *G. cambogia* (kudam puli, GC1–GC5) of Indian origin were acquired online. The samples were freeze-dried, ground to fine particles using a domestic mill (Moulinex, Spain), and sieved through a 500 µm mesh. Samples were stored in dry and hermetically closed recipients protected from light at ambient temperature until extraction.

*G. cambogia* food supplements (GCFS1–GCFS17) were acquired either online or in specialized shops, pharmacies, and supermarkets. Appendix A of Supplementary Material shows the GCFS formulation and composition declared in each label. GCFS1A and GCFS1B were supplements from the same brand but of different production batches. Three formulation units per supplement were randomly selected, crushed in a porcelain mortar (in the case of tablets), homogenized, and sieved through an Advantech No. 35 sieve (Taipei, China) of a 500 μm mesh size. All samples were stored in a cool and dry environment.

### 2.3. Reference Fruit Extracts

Reference extracts were obtained from the freeze-dried fruit samples GC1–GC5 by solid–liquid extraction (SLE) under magnetic stirring. Different extraction temperatures (25, 45, and 100 °C) and times (30 and 60 min) were evaluated in terms of HCA concentration using 0.25 g of GC1 and 10 mL of milli-Q water. After extraction, samples were centrifuged at 4930× *g* for 10 min and the supernatant was collected, filtered through 0.22 µm, and kept at −20 °C until analysis. All reference extracts were obtained by duplicate.

### 2.4. Gas Chromatography–Mass Spectrometry

Before GC–MS analyses, 400 mg of each supplement was dissolved in 5 mL of milli-Q water. After stirring for 20 min at room temperature, samples were filtered through 0.22 µm and kept at 4 °C until analysis. All samples were prepared in duplicate. *G. cambogia* fruit extracts (0.2 mL) or supplement solutions (0.3 mL) were mixed with 0.10 mL of internal standard (phenyl *β*-D-glucopyranoside) solution (1 mg mL^−1^) and evaporated under vacuum (miVac Duo concentrator, Inycom, Madrid, Spain) at 37 °C.

The formation of trimethylsilyl oximes (TMSO) was selected for GC–MS analyses. This procedure allows for reducing the number of chromatographic peaks of reducing sugars that could be obtained with silylated derivatives (up to 5 peaks corresponding to the *α* and *β* pyranose, *α* and *β* furanose forms, and the open chain) to only two peaks (corresponding to the *E* and *Z* forms). Derivatization was carried out as follows: (i) Oximation: 2.5% hydroxylamine chloride solution in anhydrous pyridine (350 µL) was added, stirred, and heated at 75 °C for 15 min; (ii) Silylation: HMDS (350 µL) and TFA (35 µL) were added, stirred, and heated at 45 °C for 30 min. The derivatized samples were centrifuged in a MiniSpin 5452000018 microcentrifuge (Eppendorf, Hamburg, Germany) for 10 min at 4401× *g*. The supernatant was collected and analyzed by GC–MS.

Derivatized samples and standards were analyzed in a 6890 plus gas chromatograph coupled to a single-quadrupole 5973 network mass spectrometer, both from Agilent Technologies (Palo Alto, CA, USA). Chromatographic analyses were carried out on an SGE HT-5 polycarborane capillary column (30 m × 0.22 mm i.d., 0.10 µm film thickness; SGE, Melbourne, Australia), using helium as the carrier gas.

The oven temperature was programmed at 150 °C for 5 min; the temperature was then increased by 15 °C min^−1^ to 340 °C and was held for 6 min. Injections (1 µL) were carried out in split mode (1:20) at a temperature of 300 °C. The mass spectrometer transfer line was set at 280 °C and ion source at 230 °C with the electron ionization mode set at 70 eV, and the scan mode range was set to 50–650 *m*/*z*.

Eluting compounds were identified using available analytical standards (Section 2.1) and data from spectral libraries (Wiley, NIST). Quantitative analysis was carried out using the internal standard method based on the preparation of calibration curves (0.01–0.5 mg mL^−1^) of *myo*-inositol, glucose, fructose, and verbascose. These calibration curves were also used for the quantitation of isomers or related compounds (*myo*-inositol for *muco*-inositol; glucose for galactose, pentitols, arabitol, and arabinose; fructose for mannitol; raffinose for maltotriose; and verbascose for maltotetraose). A response factor equal to 1 was assumed for DFAs, citric acid, vitamin B6, and vitamin C.

### 2.5. High-Performance Liquid Chromatography Analysis

Previous to the HPLC analysis, 5 mg of GCFS was dissolved in 5 mL of 8 mM H_2_SO_4_ for 20 min at 25 °C under constant stirring. Finally, samples were filtered through 0.22 µm nylon filters (Chromlab S.L., Barcelona, Spain) and analyzed via LC–UV. All analyses were carried out in triplicate.

Analyses were performed on an Agilent 1200 series HPLC system (Hewlett-Packard, Palo Alto, CA, USA) equipped with an oven and a UV–vis detector following the method developed by Jayaprakasha and Sakariah [21] with some modifications. Samples (20 µL) were injected using a Rheodyne 7725 valve, the UV wavelength was set at 210 nm, and a C_18_ analytical column (250 mm × 4.6 mm, 5 μm; Teknokroma, Barcelona, Spain) was used. The mobile phase consisted of 8 mM H_2_SO_4_ (solvent A) and acetonitrile (solvent B) with a flow rate of 1 mL min^−1^ and a linear gradient as follows: 0 min, 0% B; 7 min, 0% B; 10 min, 7% B; 15 min, 100% B; and 25 min, 100% B. After this, initial conditions were resumed and held constant for 10 min to re-equilibrate the column. HP Chemstation software was used for data acquisition. The identification of (-)-HCA and (-)-HCAL was carried out by comparing the retention times of the analytical standards with those found in the samples. Quantitation analyses were carried out by the external standard method, using calibration curves of HCA standard solutions from 0.05 to 1 mg mL^−1^ in 8 mM H_2_SO_4_. All analyses were carried out in triplicate.

Different parameters were considered for the validation of the method: recovery, precision, linearity, limit of detection (*LOD*), and limit of quantitation (*LOQ*). The recovery was evaluated by adding three different concentrations of (-)-HCA standard (0.05 mg mL^−1^, 0.15 mg mL^−1^, and 0.25 mg mL^−1^; *n* = 3) to GCFS17 sample. The precision of the method was measured on the basis of repeatability precision and intermediate precision by analyzing (-)-HCA standard at different concentration levels (0.05 to 1 mg mL^−1^) within the same day (*n* = 5) and on 5 different days, respectively. Linearity was evaluated at different concentrations of (-)-HCA standard (0.05 to 1 mg mL^−1^ range). The goodness of fit of the calibration curves was evaluated using their correlation coefficients. The *LOD* and *LOQ* were calculated for (-)-HCA as three and ten times the signal-to-noise ratio (S/N), respectively. Validation data can be found in Appendix A.

### 2.6. Statistical Analysis

Statistical analysis was carried out using the Statistica 7.0 program (StatSoft, Inc. Tulsa, OK, USA). The compliance between the experimental and declared values of HCA and the HCA concentrations of kudam puli extracts under different conditions were assessed by *t*-test (*p* < 0.01).

## 3. Results and Discussion

### 3.1. Analysis of Garcinia Fruit Rind Extracts

Different temperatures and times were considered to obtain reference extracts from kudam puli samples. Regarding temperature, the greatest extraction of HCA was obtained at 45 °C (20 ± 2%) and 100 °C (19.3 ± 0.3%), while significantly lower concentrations were found at 25 °C. Moreover, there were no significant differences between the HCA contents extracted at 30 and 60 min. From these results, extracts of *G. cambogia* fruit rinds (GC1–GC5) were obtained at 45 °C for 30 min and were analyzed via LC–UV and GC–MS to obtain a genuineness profile that could be compared with those obtained for GCFS.

Figure 1A shows the LC–UV profile of GC1 kudam puli extract. (-)-HCA, (-)-HCAL, and citric acid were detected in all the samples, as previously reported by other authors [14,20,26,27]. Despite the low polarity and small size of these acids, an acceptable resolution between them was achieved under the applied chromatographic conditions. Figure 1B shows the GC–MS chromatogram obtained for sample GC1, previously subjected to the derivatization process described in Section 2.4. In general, similar profiles were observed for all the fruit rinds analyzed. The monosaccharides glucose and fructose were the main sugars detected, while other carbohydrates such as arabinose, galactose, and *myo*-inositol were also found at lower levels. Among the organic acids, (-)-HCA, (-)-HCAL, and citric acid were also detected. Moreover, arabinoic acid γ lactone was found in some samples. Other minor peaks containing a 292 *m*/*z* ion, characteristic of polyhydroxy carboxilic acids [28], and a compound (t_R_: 9.63 min) with a mass fragmentation pattern similar to that of (-)-HCA (*m*/*z* 73, 115, 147, 292, and 259 in Appendix A) were also detected. These peaks could correspond to compounds produced by the degradation of HCA or HCAL during the derivatization process. Other authors have shown that due to the highly hygroscopic nature of HCA, incomplete derivatization reactions can be produced [20]. Thus, LC–UV was selected for the quantitative analysis of these acids, while other compounds of kudam puli extracts were quantified by GC–MS.

Table 1 shows the concentrations (mg g^−1^) of the different compounds found in kudam puli fruit extracts determined by LC–UV or GC–MS. The (-)-HCA content varied from 77 to 121 mg g^−1^, while (-)-HCAL was present in the range of 93 to 140 mg g^−1^, with GC4 being the sample that showed the highest values for both compounds. These values agreed with those found by other authors in *Garcinia cambogia* rinds (between 10% and 30% of (-)-HCA in the free form or as its lactone) [14,20].

In general, glucose and fructose were the most abundant carbohydrates in kudam puli extracts. A great variability in their concentrations was observed among samples; glucose concentrations ranged between 3.9 and 38 mg g^−1^ fruit, while those of fructose ranged between 3.5 and 31 mg g^−1^ fruit. These differences may be attributed to different origins of the plant, maturation of the fruit, processing, or storage conditions [29]. Although the variability in glucose and fructose concentrations was high for the different samples, in general, glucose/fructose ratios (G/F) presented smaller differences (between 0.9 and 1.5). Slight variations were also found for *myo*-inositol concentrations (values between 0.11 and 0.16 mg g^−1^, except for GC1, which presented a greater concentration).

Few references can be found in the literature reporting the presence of carbohydrates in *G. cambogia* fruit rinds [14,25,30]; however, to the best of our knowledge, no information about the qualitative and quantitative composition of the carbohydrate fraction present in this matrix is available. Thus, this work contributes to the characterization of *G. cambogia* fruit rinds and allows for obtaining a genuineness profile of this natural product consumed worldwide.

### 3.2. Analysis of Garcinia Cambogia Food Supplements

Contrary to what was observed for reference samples, (-)-HCAL was not detected in the GCFS analyzed by LC–UV, and only a main peak corresponding to (-)-HCA was observed. This agrees with a report by Jayaprakasha and Sakariah [21], who analyzed four commercial *G. cambogia* extracts by LC–UV, detecting (-)-HCA as the major compound (51–55%) along with three minor peaks corresponding to tartaric, malic, and citric acids, but (-)-HCAL was not reported. Seethapathy et al. [25] only found quantifiable amounts of (-)-HCAL (*LOQ* values of 0.20 mg mL^−1^) in one of ten Garcinia food supplements analyzed by NMR. Although the sensitivity of the LC–UV methodology used in the present work was higher (*LOQ* values of 0.45 µg mL^−1^, Appendix A), this compound was not detected in any of the GCFS analyzed.

Table 2 shows the percentages of (-)-HCA determined by LC–UV and the values declared on the labels for the 18 supplements analyzed. (-)-HCA contents ranged from 4.29% in GCFS14 to 65.8% in GCFS17, and, in general, the concentrations experimentally determined were in good agreement with those stated on the labels for this acid. However, some discrepancies were detected. Samples GCFS1A, GCFS1B (obtained from the same manufacturer but from different batches), and GCFS16 presented (-)-HCA concentrations significantly lower than those declared, while higher experimental values were found for GCFS5 and GCFS15. These differences could be attributed to, among other things, the use of less accurate and selective methodologies for their quantitation (providing under- or overestimation), fraudulent practices to increase economic benefit, or inappropriate manufacturing practices during product formulation.

The concentrations of (-)-HCA in GCFS13 and GCFS14 were not indicated on their labels, but it is worth noting the low values experimentally detected for the active ingredient (6.57 and 4.29%, respectively). These percentages correspond to daily intakes of 67 mg and 49 mg of HCA, which are extremely low, considering that the dosages of (-)-HCA reported in clinical trials to exert a beneficial effect ranged from 900 to 2800 mg/day [31]. This brings into question the efficiency of these GCFS for overweight control.

Analysis by GC–MS allowed the detection of other compounds present in the *G. cambogia* supplements (Figure 2 and Table 3 and Table 4). The qualitative and quantitative profiles were compared with those obtained for the reference samples. In general, three trends could be observed for compounds other than (-)-HCA in the GCFS analyzed: (i) food supplements composed of *myo*-inositol and small amounts of sugars (Figure 2A; GCFS: 1A, 1B, 3, 6, 7, 11, 13, 14, and 17); (ii) those GCFS that contained large amounts of carbohydrates, mainly maltodextrins (Figure 2B; GCFS: 2, 4, 5, 8, and 12); and finally, (iii) supplements constituted only by small concentrations of *myo*-inositol (Figure 2C; GCFS: 9, 10, 15, and 16). 

Carbohydrates such as glucose, fructose, arabinose, galactose, and *myo*-inositol, present as natural components of the *G. cambogia* reference samples, were detected in most of the food supplements (groups (i) and (ii)). The presence of *myo*-inositol in all the FSOC analyzed should be highlighted. The highest concentrations of this cyclitol were found in GCFS5 (0.24 mg g^−1^) and GCFS17 (0.23 mg g^−1^). It has been described that *myo*-inositol is a polyol present in plasma membranes and in other structures of natural products, including fruits and vegetables [32,33,34]. The presence of this compound could be used as an indicator of the natural origin of these supplements. It is worth noting that this compound is also present in samples from group (iii), which were probably subjected to exhaustive purification of (-)-HCA.

Maltose, maltotriose, and maltotetraose were detected in high amounts in some of the GCFS analyzed (group (ii)). These sugars, absent in the reference extracts, could come from excipients added to the supplements, such as maltodextrins, commonly used as bulking and spray-drying agents to allow their adequate manufacture [35]. The variability observed with respect to the relative abundances of maltose, maltotriose, and maltotetraose in the different samples could be due to the different degree of hydrolysis of the maltodextrins used in their production (the higher the degree of hydrolysis, the shorter the glucose chains). In those cases with a lower degree of maltodextrin hydrolysis and containing glucose molecules with a high degree of polymerization (DP > 4), they would not be detected by GC–MS. This could be the case for GCFS11, which declared maltodextrins in its label, but none of these carbohydrates were detected. In general, 70% of the analyzed FSOC containing maltodextrins did not declare their content on their labels (Appendix A). Although the use of maltodextrins as food additives is approved by regulatory agencies [36], these carbohydrates could generate an unwanted caloric intake in supplements, especially in those such as GCFS, intended to control overweight. This is particularly important in those supplements for which concentration of maltodextrins is high. A clear example is the supplement GCFS2, with the highest concentrations of both maltose and maltotriose (16.85 and 153.4 mg g^−1^, respectively) and high amounts of maltotetraose (17 mg g^−1^).

High variation was also observed in glucose and fructose concentrations. The highest values of glucose were observed in the GCFS4 sample, with 2.25 mg g^−1^, while GCFS11 and GCFS17 showed the lowest values for this sugar (0.015 and 0.017 mg g^−1^, respectively). The high amount of glucose may be due to the addition of maltodextrins as excipients, as occurred with the presence of maltose and maltotriose. Regarding fructose, GCFS3, GCFS6, and GCFS4 showed concentrations of 2.4, 1.9, and 1.5 mg g^−1^, values noticeably higher than those for the remaining supplements. High concentrations of fructose in GCFS could be attributed to the use of high-fructose syrup as an additive, a sweetener widely used in the food industry. These differences in glucose and fructose concentrations gave rise to wide variation in the G/F ratios compared with those of reference extracts. The G/F ratios of FSOC varied from 0.05 (GCFS3) to 22.1 (GCFS12); only supplements GCFS1B, GCFS4, and GCFS11 showed values within the G/F range found for the reference extracts.

Apart from (-)-HCA, other acids such as citric, ascorbic, gluconic, and glucaric acids were also detected in some of the samples. Citric acid, which was found to be present in most of the GCFS analyzed, was also found in the reference samples. GCFS4 showed a high concentration of ascorbic acid (72 mg g^−1^), which was declared as an additive on its label.

In some supplements, the presence of difructose anhydrides (DFAs) was also observed. DFAs are non-fermentable pseudodisaccharides, produced by the condensation of two fructose molecules [37]. They are formed by subjecting foods rich in reducing sugars, such as high-fructose corn syrups or inverted sugar syrups, to heat treatments [38,39]. Therefore, their presence in *G. cambogia* supplements could be due to processing or storage conditions applied during their manufacture or to the addition of fructose syrup to potentiate their flavor. It must be pointed out that the presence of DFAs was detected in all samples with high fructose contents, probably coming from the addition of fructose syrups.

Pinitol (methyl-*chiro*-inositol; *t_R_*: 6.78 min) and *muco*-inositol (*t_R_*: 7.65 min) were detected in GCFS17 at low concentrations; however, they were not found in reference kudam puli extracts. Their presence could be attributed to the contribution of other natural sources added to this supplement. Both polyalcohols have been previously detected in honeys [40]; however, their presence in GCFS17 could not be attributed to this food since other carbohydrates typical of honey were not detected.

As reported on their labels, the presence of mannitol was confirmed in GCFS7, as were vitamin B6 and vitamin C in the GCFS1A and GCFS1B supplements. This last compound was also detected in GCFS4, as indicated on its label. Galactosyl-glycerol (*m*/*z* 204, 217, and 337) and digalactosyl-glycerol (*m*/*z* 204, 337, and 597) were also identified in some supplements. These compounds, not present in the reference samples, have been detected in several natural sources such as cocoa beans [33], mung bean [41], and kale [33].

## 4. Conclusions

GC–MS and LC–UV analyses allowed the qualitative and quantitative characterization of *G. cambogia* fruit rind extracts, providing an authenticity profile that could be compared with those of GCFS. The proposed multi-analytical methodology proved to be successful for the detection of compounds in GCFS that were not present in the natural source, as well as others probably coming from the undeclared addition of different additives that could affect their nutritional properties. Some of them, such as maltodextrins, can generate an unwanted caloric intake in supplements intended to control overweight, while others can provide beneficial effects, such as vitamins. In addition, discrepancies were observed in the declared content of bioactive compounds ((-)-HCA). This study demonstrates the potential of the multi-analytical methodology for the quality evaluation of commercial *G. cambogia* food supplements, assuring the composition in bioactives, and detecting possible fraudulent practices that would be of great interest to regulatory authorities.

## Figures and Tables

**Figure 1 nutrients-14-03077-f001:**
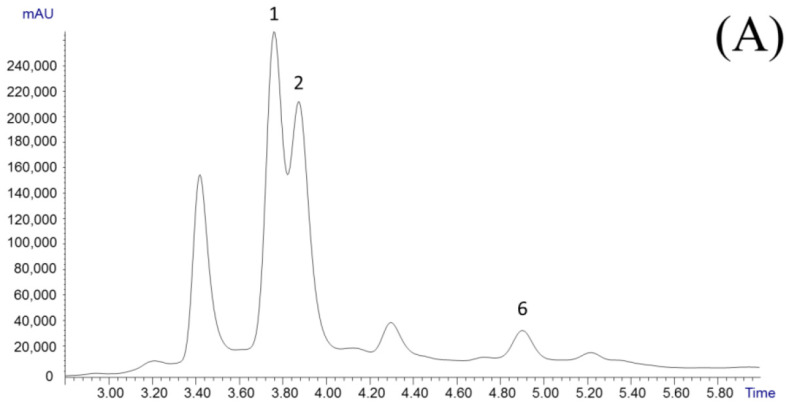
Dried *G. cambogia* chromatograms obtained for GC1 extract by LC–UV (**A**) and GC–MS (**B**). Peak assignations shown in Table 1. * Unknown. I.S.: Internal standard.

**Figure 2 nutrients-14-03077-f002:**
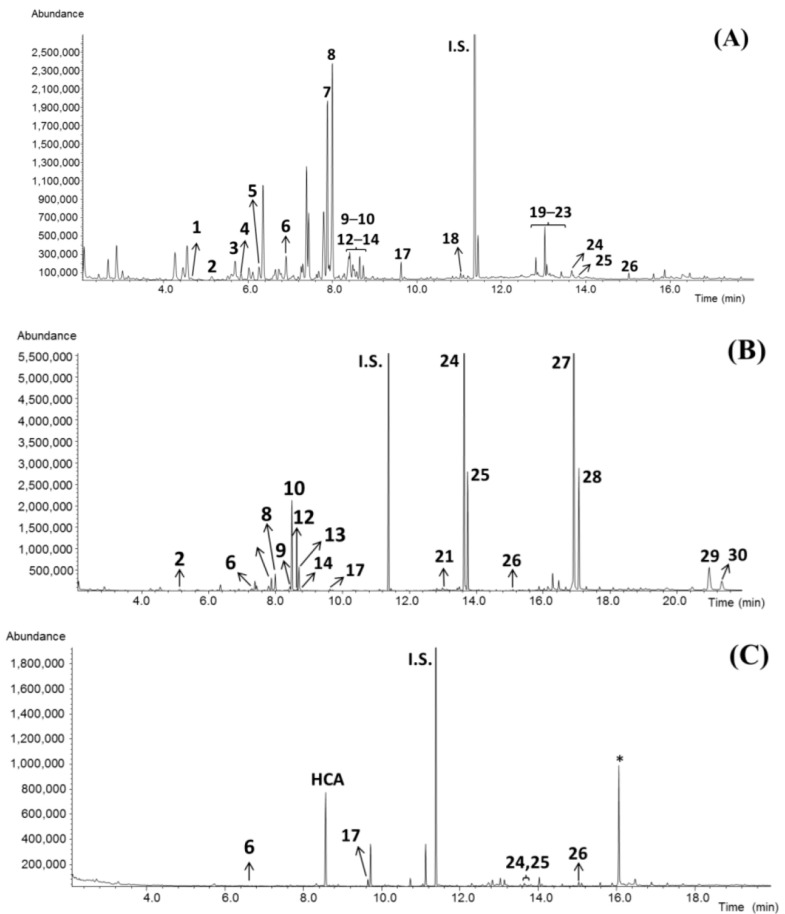
GC–MS chromatograms obtained for samples GCFS3 (**A**), GCFS8 (**B**), and GCFS9 (**C**). Peak assignations shown in Table 3 and Table 4. * Artifact.

**Table 1 nutrients-14-03077-t001:** Composition (mg g^−1^) of kudam puli extracts determined by LC–UV and GC–MS (*n* = 3).

	Samples					
Compound (mg g^−1^)	ID	GC1	GC2	GC3	GC4	GC5
(-)-HCAL ^#^	1	108 (5)	93 (18)	115 (20)	140 (31)	99 (22)
(-)-HCA ^#^	2	87.7 (0.6)	77 (13)	107 (21)	121 (29)	89 (14)
Arabinonic acid γ lactone ^$^	3	0.28 (0.03)	0.26 (0.01)	-	-	0.45 (0.02)
Arabinose ^$^	4	22 (4)	22. 40 (0.02)	15 (1)	20 (1)	11.5 (0.8)
Pentitol ^$^	5	0.20 (0.08)	tr	tr	tr	tr
Citric acid ^$^	6	6 (1)	4.8 (0.4)	18 (2)	10.2 (0.3)	5.5 (0.6)
Fructose (F) ^$^	7	25 (3)	20.4 (0.6)	31 (4)	8.02 (0.03)	3.5 (0.1)
Galactose ^$^	8	0.37 (0.06)	0.34 (0.03)	0.3 (0.1)	0.82 (0.02)	1.34 (0.06)
Glucose (G) ^$^	9	38 (4)	27.01 (0.02)	33 (4)	7.2 (0.2)	3.9 (0.3)
*myo*-inositol ^$^	10	0.43 (0.02)	0.1336 (0.0004)	0.15 (0.02)	0.16 (0.02)	0.106 (0.001)
G/F	-	1.54 (0.02)	1.32 (0.04)	1.070 (0.005)	0.86 (0.02)	1.13 (0.06)

^#^: Determined by HPLC–UV. ^$^: Determined by GC–MS.

**Table 2 nutrients-14-03077-t002:** Percentages of (-)-HCA experimentally determined in the FSOC analyzed and declared on their labels.

	(-)-HCA (%, *w*/*w*)	
FSOC	Experimental *	Declared
GCFS1A	21.35 (2.01) ^b^	47 ^a^
GCFS1B	19.80 (0.98) ^b^	47 ^a^
GCFS2	33.70 (1.08) ^a^	34 ^a^ 47 ^a^
GCFS3	52.12 (2.54) ^a^	
GCFS4	30.31 (7.47) ^a^	36 ^a^
GCFS5	56.00 (1.62) ^a^	46 ^b^
GCFS6	54.60 (7.42) ^a^	56 ^a^
GCFS7	42.86 (2.09) ^a^	41 ^a^
GCFS8	49.81 (2.71) ^a^	51 ^a^
GCFS9	50.27 (3.81) ^a^	48 ^a^
GCFS10	46.62 (2.38) ^a^	48 ^a^
GCFS11	10.45 (0.28) ^a^	12 ^a^
GCFS12	46.42 (1.42) ^a^	48 ^a^
GCFS13	6.57 (0.62)	-
GCFS14	4.29 (0.84)	-
GCFS15	60.13 (1.05) ^a^	55 ^b^
GCFS16	46.91 (6.49) ^b^	60 ^a^
GCFS17	65.8 (2.1) ^a^	62 ^a^

* Content calculated based on the declared extract of *G. cambogia* on the label. ^a,b^ Means statistically different from the theoretical value (declared) at a *p* < 0.01 probability level.

**Table 3 nutrients-14-03077-t003:** Concentrations of carbohydrates (mg g^−1^) found in *Garcinia* food supplements (GCFS).

ID.	Compound	1A	1B	2	3	4 ^&^	5	6	7 ^$^	8	9	10	11	12	13	14	15	16	17 ^#^
1	Pentitol	-	-	-	0.025 (0.006)	-	-	-	0.03 (0.01)	-	-	-	-	-	-	-	-	-	tr
2	Arabitol	0.040 (0.007)	0.033 (0.003)	0.018 (0.010)	0.049 (0.003)	tr	0.663 (0.001)	0.075 (0.002)	0.046 (0.002)	0.0344 (0.0005)	-	0.11 (0.05)	-	-	tr	0.07 (0.02)	-	-	-
3, 4	Arabinose	-	-	0.101 (0.006)	0.22 (0.02)	0.5 (0.2)	0.184 (0.002)	0.9 (0.2)	0.5 (0.1)	-	-	-	0.0191 (0.0003)	-	0.02 (0.01)	-	-	-	0.32 (0.04)
5	Pentitol	0.048 (0.001)	0.042 (0.001)	0.055 (0.001)	0.114 (0.002)	-	0.0611 (0.0008)	0.18 (0.04)	0.17 (0.05)	-	-	-	-	-	-	-	-	0.024 (0.003)	0.076 (0.001)
7, 8	Fructose (F)	0.0842 (0.0002)	0.13 (0.06)	0.08 (0.03)	2.4 (0.1)	1.48 (0.03)	0.271 (0.003)	1.9 (0.3)	0.6 (0.2)	0.335 (0.002)	-	-	0.0145 (0.0001)	0.0169 (0.0004)	0.005 (0.003)	-	-	-	0.034 (0.006)
9, 14	Galactose	tr *	0.038 (0.007)	0.08 (0.02)	0.35 (0.03)	0.44 (0.02)	0.28 (0.04)	0.15 (0.03)	0.04 (0.01)	0.072 (0.003)	-	-	-	-	0.005 (0.002)	0.022 (0.001)	-	-	0.026 (0.003)
10, 13	Glucose (G)	0.187 (0.043)	0.172 (0.008)	0.491 (0.004)	0.11 (0.02)	2.25 (0.08)	0.55 (0.02)	0.38 (0.10)	0.075 (0.007)	1.18 (0.02)	-	-	0.015 (0.002)	0.373 (0.004)	0.021 (0.005)	0.144 (0.006)	-	-	0.017 (0.002)
17	*myo*-inositol	0.025 (0.002)	0.022 (0.001)	0.040 (0.001)	0.11 (0.02)	0.086 (0.005)	0.24 (0.02)	0.064 (0.005)	0.056 (0.002)	0.02 (0.01)	0.023 (0.005)	0.04 (0.02)	0.005 (0.001)	0.0163 (0.0005)	0.013 (0.004)	0.022 (0.002)	tr	0.023 (0.003)	0.23 (0.03)
18	Galactosyl-glycerol	-	-	-	0.018 (0.001)	-	0.042 (0.007)	-	-	-	-	-	-	-	-	-	-	-	-
19–23	DFAs	-	-	0.08 (0.03)	0.51 (0.03)	0.13 (0.04)	0.20 (0.04)	0.30 (0.05)	0.15 (0.03)	0.030 (0.004)	-	-	tr	-	0.045 (0.005)	0.05 (0.02)	-	-	-
24, 25	Maltose	0.104 (0.002)	0.0855 (0.0001)	16.85 (0.01)	0.063 (0.001)	15.3 (1.3)	2.2 (0.1)	-	tr	6.9 (0.1)	tr	-	-	5.3 (0.3)	0.6 (0.1)	-	-	-	-
26	Digalactosyl-glycerol	0.008 (0.001)	0.007 (0.003)	0.0108 (0.0004)	0.0281 (0.0003)	0.025 (0.003)	0.02 (0.01)	0.012 (0.008)	0.006 (0.003)	0.004 (0.002)	0.025 (0.004)	tr	-	0.016 (0.003)	-	-	-	-	0.0113 (0.0003)
27, 28	Maltotriose	0.381 (0.008)	tr	153.4 (15.4)	-	14.2 (183)	11.6 (0.4)	-	-	41.9 (0.9)	-	-	-	27.8 (4.4)	-	-	-	-	-
29, 30	Maltotetraose	-	-	17.0 (2.2)	-	2.3 (0.2)	10.3 (0.1)	-	-	33.6 (1.86)	-	-	-	0.9 (0.1)	-	-	-	-	-
-	G/F	2.2 (0.5)	1.5 (0.6)	7 (3)	0.05 (0.01)	1.53 (0.09)	2.04 (0.06)	0.20 (0.01)	0.14 (0.03)	3.50 (0.04)	-	-	1.1 (0.1)	22.1 (0.7)	4 (1)	-	-	-	0.5 (0.2)

^&^ GCFS4 contained sucrose: 0.04 (0.01) mg g^−1^; ^$^ Mannitol was detected in GCFS7: 41.4 (0.2) mg g^−1^; ^#^ GCFS17 also contained pinitol: 0.032 (0.002) mg g^−1^ and *muco*-inositol: 0.033 (0.005); Standard deviations in parentheses (*n* = 2). * tr: traces.

**Table 4 nutrients-14-03077-t004:** Concentrations of vitamins and acids (mg g^−1^) found in *Garcinia* food supplements (GCFS).

ID.	Compound	1A	1B	2	3	4	5	6	7	8	9	10	11	12	13	14	15	16	17
6	Citric acid	9.0 (0.2)	6.9 (0.6)	3.500 (0.001)	8.0 (0.4)	6.4 (0.2)	5.6 (0.2)	7.66 (0.04)	5.47 (0.05)	3.17 (0.06)	1.09 (0. 05)	4.1 (0.1)	0.677 (0.008)	1.47 (0.03)	1.5 (0.6)	-	4.1 (0.3)	2.28 (0.10)	1.06 (0.01)
11	Vitamin B6	0.177 (0.068)	0.118 (0.016)	-	-	-	-	-	-	-	-	-	-	-	-	-	-	-	-
12	Gluconic acid	0.020 (0.002)	0.019 (0.006)	0.124 (0.006)	0.16 (0.03)	21 (5)	0.066 (0.008)	0.063 (0.006)	0.053 (0.006)	0.03 (0.01)	-	0.12 (0.05)	0.01029 (0.00003)	-	0.006 (0.004)	tr	tr	-	0.031 (0.006)
15	Glucaric acid	-	-	-	-	-	0.015 (0.004)	0.0091 (0.0004)	0.011 (0.002)	-	-	-	-	-	-	-	-	tr	-
16	Vitamin C	6.02 (0.01)	5.7 (0.1)	-	-	72.2 (0.1)	-	-	-	-	-	-	-	-	-	-	-	-	-

Standard deviations in parentheses (*n* = 2).

## Data Availability

The data presented in this study are available on request from the corresponding author.

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
