# Peer review of "Quality Evaluation of Dietary Supplements for Weight Loss Based on Garcinia cambogia"

_nutrients, 2022, doi:10.3390/nu14153077_

Round 1
Reviewer 1 Report
The reviewer appreciate the interest of the authors in the development of this manuscript. It is an interesting topic.
1. The main objective of this study was to determine the quality of dietary supplements containing G. cambogia by applying LC-UV and GC-MS methods for qualitative and quantitative analyses of (-)-hydroxycitric acid ((-)-HCA), as well as other components of these supplements that may affect product quality.
Line: 33-36. Please this statement needs a bibliographic reference
Line: 52-54. Please this statement needs a bibliographic reference.
Line: 72-73. Please this statement needs a bibliographic reference.
Line: 111-116. Did the authors optimize the extraction method?
Line: 215-216. Please this statement needs a bibliographic reference.
The authors do not state validation parameters presented methods.
The paper lacks an answer to the question: What is the nutritional significance of comparing, in terms of composition, a natural G. cambogia product with a dietary supplement?
The authors mention the use of unfair practices by manufacturers. They particularly mention this in the case of the assessment of the (-)-HCA content, the labelled value of which is higher than that declared by the manufacturer. To confirm this statement, the authors should provide the validation parameters of the method used. Without this information, the authors' assumptions may be questioned.
The reviewer appreciate the interest of the authors in the development of this manuscript. It is an interesting topic. However, the work should be supplemented with the information listed. I suggest MAJOR REVISIONS.
Author Response
Line: 33-36. Please this statement needs a bibliographic reference
Line: 52-54. Please this statement needs a bibliographic reference.
Line: 72-73. Please this statement needs a bibliographic reference.
All these references have been included.
Line: 111-116. Did the authors optimize the extraction method?
The extraction method was optimized to obtain the highest HCA concentrations. Optimization conditions and results have been now included in materials and methods and results and discussion sections.
Line: 215-216. Please this statement needs a bibliographic reference.
Reference has been included.
The authors do not state validation parameters presented methods.
Validation parameters of HCA method have been now included in the manuscript as supplementary table S2. GC-MS method has been previously validated and widely applied to different samples, including food supplements (Jimenez et al., 2022), thus its validation has not been reported in this manuscript.
The paper lacks an answer to the question: What is the nutritional significance of comparing, in terms of composition, a natural G. cambogia product with a dietary supplement?
The nutritional properties are related to the composition of a product and they can change depending on the treatment to which the sample is subjected. Then, in the case of G. cambogia supplements, their nutritional significance would be related to their content in HCA but also in other constituents of G. cambogia fruits which are preserved in the supplements (myo-inositol, carbohydrates…). Moreover, as it is described in results and discussion section, other compounds are intentionally added in the manufacture process of supplements, which provide nutritional added value to these products, such as vitamins. However, other compounds, such as maltodextrins, could generate an unwanted caloric intake, especially in those food supplements intended to control overweight. This is particularly important in those supplements for which concentration of maltodextrins is high. It has been commented in the results and discussion and in the conclusion sections.
The authors mention the use of unfair practices by manufacturers. They particularly mention this in the case of the assessment of the (-)-HCA content, the labelled value of which is higher than that declared by the manufacturer. To confirm this statement, the authors should provide the validation parameters of the method used. Without this information, the authors' assumptions may be questioned.
The validation parameters of the method used to determine HCA are now included in the manuscript.
Reviewer 2 Report
The manuscript is well-written and presents interesting data about the composition of nutritional supplements containing G. cambogia extracts already present on the market. The topic is hot, since nutraceuticals are gaining a lot of interest during the last years, and represent dominant products in the market of those destined to wellbeing. For this reason, more restricted controls concerning the chemical composition of nutraceuticals are required, in order to avoid the entrance in the market of useless or potentially harmful products. In this view, the work here presented shows important results.
However, some minor issues were found, and they are listed below. In my opinion, after checking and implementing some metodological aspects, the work can be accepted for publication in Nutrients.
Methods
Lines 112-13: Was extraction performed by maceration or was it implemented by using ultrasound? Same for Lies 118-20.
Lines 145-47: tentative identification in GC-MS should be performed by considering the retention indeces (or Kovats incdeces) of compounds. From what is reported in these lines, it is not clear if these values were considered or if only the retention time was used. In this last case, how was retention time used for identification?
For both LC-MS and GC-MS, please report how LODs and LOQs were calculated. Furthermore, were the quantitative methods used in this study validated? Also validation critera should be reported.
Results
All the Figures should be reported in higher resolution (e.g., 300x300 DPI). Furthermore, Figure 1A should be checked, since the peak at 4.3 min is cut on the top.
Line 191: Which acids?
Table 2. Please, report the w/w% also in the “declared” column. Also, indicate what the apices “a” and “b” stand for.
Table 3 should be reformulated, maybe splitted in two tables.
Author Response
Lines 112-13: Was extraction performed by maceration or was it implemented by using ultrasound? Same for Lies 118-20.
Extraction was carried out by solid-liquid extraction (SLE); ultrasound was not used. It has been now clarified in materials and methods.
Lines 145-47: tentative identification in GC-MS should be performed by considering the retention indeces (or Kovats incdeces) of compounds. From what is reported in these lines, it is not clear if these values were considered or if only the retention time was used. In this last case, how was retention time used for identification?
Referee is right. Linear retention indices should be calculated to identify unknown compounds, however, in this case, most of the compounds found by GC-MS were identified by comparison of their retention times and mass spectrum with those of analytical standards. In particular, pentitols could not be assigned and the general term was used; moreover, vitamin B6 was also tentatively assigned using the Wiley library. Text in the manuscript has been changed.
For both LC-MS and GC-MS, please report how LODs and LOQs were calculated. Furthermore, were the quantitative methods used in this study validated? Also validation critera should be reported.
Validation parameters of LC-UV methods have been indicated in the revised version of the manuscript (including LOD and LOQ calculation procedure). GC-MS method has been previously validated and widely applied to different samples, including food supplements (Jimenez et al., 2022), thus its validation has not been reported in this manuscript.
All the Figures should be reported in higher resolution (e.g., 300x300 DPI). Furthermore, Figure 1A should be checked, since the peak at 4.3 min is cut on the top.
Figures have been improved and included in the text. Moreover, figures are attached independently.
Line 191: Which acids?
This fragment is characteristic of polyhydroxy carboxilic acids. It has been changed and a reference has been included.
Table 2. Please, report the w/w% also in the “declared” column. Also, indicate what the apices “a” and “b” stand for.
Changes have been done.
Table 3 should be reformulated, maybe splitted in two tables.
Table 3 has been adjusted to try to do it clearer. In our opinion, is more visual to report all these data in a single table, but if the editor considers it appropriate we can divide it as attached.
Round 2
Reviewer 1 Report
After the authors have incorporated the suggested revisions to the manuscript, I recommend the paper for publication in the journal.